# Meropenem/Vaborbactam: β-Lactam/β-Lactamase Inhibitor Combination, the Future in Eradicating Multidrug Resistance

**DOI:** 10.3390/antibiotics12111612

**Published:** 2023-11-10

**Authors:** Anna Duda-Madej, Szymon Viscardi, Ewa Topola

**Affiliations:** 1Department of Microbiology, Faculty of Medicine, Wroclaw Medical University, Chałubińskiego 4, 50-368 Wrocław, Poland; 2Faculty of Medicine, Wroclaw Medical University, Ludwika Pasteura 1, 50-367 Wrocław, Poland; szymon.viscardi@student.umw.edu.pl (S.V.); ewa.topola@student.umw.edu.pl (E.T.)

**Keywords:** β-lactam/β-lactamase inhibitor, carbapenem-resistant *Enterobacterales* (CRE), *Klebsiella pneumoniae* carbapenemase (KPC), meropenem, multidrug resistance, vaborbactam

## Abstract

Due to the fact that there is a steadily increasing trend in the area of antimicrobial resistance in microorganisms, there is a need to look for new treatment alternatives. One of them is the search for new β-lactamase inhibitors and combining them with β-lactam antibiotics, with the aim of increasing the low-dose efficacy, as well as lowering the resistance potential of bacterial strains. This review presents the positive effect of meropenem in combination with a vaborbactam (MER-VAB). This latest antibiotic-inhibitor combination has found particular use in the treatment of infections with the etiology of carbapenem-resistant *Enterobacterales* (CRE), Gram-negative bacteria, with a high degree of resistance to available antimicrobial drugs.

## 1. Introduction

Antimicrobial resistance (AMR) is an ever-growing problem in medical care. It reduces the effectiveness of the therapy, increases the risk of death, and accelerates the spread of infections with the etiology of multidrug-resistant (MDR) bacteria. In addition, it creates the need for very powerful alternative therapies, resulting in negative side effects on a patient’s health and quality of life. This phenomenon is driven by the excessive use of antibiotics (1) in clinically unjustified cases (e.g., in viral contamination) and (2) outside the “ACCESS” list: according to the WHO (World Health Organization) AWaRe (Access, Watch, Reserve) classification, effective against a wide range of infections, and induces low resistance levels [1]. In response to such irresponsible behaviors, bacteria evolve and naturally develop resistance to drugs to which they had previously shown sensitivity.

In recent years, the antibiotic consumption has increased dramatically; hence, The Global Antimicrobial Resistance and Use Surveillance System (GLASS) has decided to introduce analyses for AMR indicators in the context of antimicrobial drugs consumption [2]. According to the collected data, the transition to the 21st century led to a 65% increase in the defined daily dose (DDD) of antimicrobial substances. In contrast, antibiotic usage increased by 39% (from 11.3 to15.7 DDD per 1000 inhabitants per day) [3]. Moreover, the consumption of DDDs in pediatric patients raises concerns. A study by Browne et al. found that the DDDs increased by 46% in 2018 (37.2–43.7 DDD per 1000 inhabitants per day) compared to 2000 (9.2–10.5 DDD per 1000 inhabitants per day) [4]. However, although these figures have been reported as the pre-COVID-19 (coronavirus disease 2019) pandemic conditions, they have already raised major concern. The post-COVID-19 era seems to break down under all the rules. The analyses conducted by Sulayyim et al. showed that the antibiotic resistance during the COVID-19-related pandemic increased among both Gram-negative (i.e., *Acinetobacter baumannii*, *Klebsiella pneumoniae*, *Escherichia coli,* and *Pseudomonas aeruginosa*) and Gram-positive (i.e., *Staphylococcus aureus* and *Enterococcus faecium*) bacteria. The observed resistance was mainly against antibiotics used in the empirical therapy of in-hospital (colistin and ciprofloxacin) and also out-of-hospital (ampicillin and erythromycin) pneumonia [5]. However, it is disturbing to note that resistance to the so-called “last chance” antibiotic (colistin) has enhanced in the absence of effectiveness of other antimicrobial agents. The results of other studies also confirmed this alarming phenomenon, even though they showed increased resistance to other antibiotics, i.e., imipenem and ceftazidime, used in the treatment of hospital-acquired pneumonia (including ventilator-treated pneumonia) with an etiology of Gram-negative bacteria of the *Enterobacteriaceae* family [6].

The resistance problem is a huge obstacle to appropriate therapy because only a few clinically available antibiotics can be used for the treatment. Moreover, AMR is proving to be one of the most common causes of death worldwide. According to data presented by Murray et al., more than 1.2 million people worldwide died in 2019 from a direct infection with an etiology of MDR bacteria. In turn, such contamination was an indirect cause of nearly 5 million deaths. The resistance problems particularly affect patients with pneumonia, blood infections, and appendicitis [7]. Hospital-acquired pneumonia (HAP) is one of the most common infections occurring in the hospital environment with a high potential for mortality [8,9,10]. However, although the era of antibiotics may still be in its golden age, this view is at a high risk of change. This situation, which had a protective effect for many years, could become a killer of the population. It may come into a complete circle and lead to situations in which the infections become deadly again. This could have catastrophic consequences all over the world.

Gram-negative bacteria show higher resistance than Gram-positive bacteria. They have various abilities to learn new ways to affect the failure of therapy and easily transfer their genetic material (including antibiotic resistance genes) to other bacteria. The World Health Organization has created a list of antibiotic-resistant “priority pathogens”, the critical group including MDR Gram-negative, carbapenem-resistant, and ESBLs-producing bacteria (extended-spectrum β-lactamases), i.e., *A. baumannii*, *P. aeruginosa*, *Klebsiella* spp., *E. coli*, *Serratia* spp., and *Proteus* spp. [11]. These bacterial strains are associated with serious, often fatal, health conditions, including bloodstream infection and pneumonia occurring in patients (1) residing in hospitals [12,13] and (2) long-term-care facilities [14], (3) requiring artificial ventilation [15], and (4) with an intravenous catheter in situ [16]. A recent study of this group of bacteria was more alarming because it showed that in the post-COVID-19 era, the resistance of the most critical group of bacteria increased dramatically, for example, levels for *E. coli* and *K. pneumoniae* on ampicillin raised to 89.6% and 98%, respectively; for *P. aeruginosa* on imipenem to 91.8%; and for *A. baumannii* to ceftazidime to 94.6%. In addition, an increment of about 30% has been shown for *K. pneumoniae* on ceftriaxone, a representative of the third-generation cephalosporins, which is the most effective antibiotic against multidrug-resistant strains [6]. These frightening data are thought to be underestimated by the inadequacy of some countries’ bacterial resistance monitoring systems. Thus, the obtained results and, therefore, the international response are not comparable to the current risk situation [17].

The cornerstone of antibiotic therapy is the largest and safest group referring to β-lactam antibiotics. They inactivate transpeptidases, carboxypeptidases, and endopeptidases involved in the synthesis of the bacterial cell wall. Carbapenems, which belong to this group of antibiotics, represent a last resort in the therapy of infections caused by Gram-negative *Enterobacteriaceae* [18]. However, since the β-lactam ring structure is sensitive to the action of bacterial enzymes, β-lactamases, which degrade and inactivate the antibiotic, inhibitors of these enzymes are a key factors. Thus far, several of them have been recognized, i.e., clavulanic acid, sulbactam, tazobactam, and avibactam. However, in many cases, they have already lost their effectiveness, as bacteria have adapted to the new conditions by producing newer and newer enzymes (including carbapenemases). This situation is a clear indication of the need to find and develop new antibiotics that include multidrug-resistant Gram-negative bacteria in their spectrum of action. To win this race against antibiotic resistance, it is important to use what is available by introducing modifications or enriching them with an additional substances. These new approaches aim to combat the intrinsic and acquired resistance of Gram-negative bacteria. A promising trend is natural bactericidal agents, i.e., bacteriophages, but also modified peptide antibiotics, peptide benzimidazoles, quorum sensing (QS) inhibitors, and metal-based antibacterial agents [19]. Moreover, one of such methods that gives hope for the future is to obtain the activity of these resistant bacteria by deactivating the mechanism responsible for the occurrence of resistance as a result of the action of antibiotic adjuvants that are β-lactamase inhibitors.

In this review, we compile the data on the combination of a β-lactam antibiotic with an inhibitor, meropenem with a vaborbactam. This latest interconnection, through its unique action, has found application in the therapy of critically ill patients, mainly in the ICU (intensive care unit). Furthermore, it has shown to be an effective therapy for treating very difficult infections with CRE (carbapenem-resistant *Enterobacterales*) etiology.

## 2. Characteristics

As the prevalence of multidrug-resistant strains increases, the effectiveness of older β-lactamase inhibitors (BLIs) used with β-lactams (BLs) decreases. Resistance to carbapenems is very important in bacteria such as *Pseudomonas* spp., *Acinetobacter* spp., and the *Enterobacterales* and is an indicator of selection for the appropriate antibiotic [20]. The increased demand for new methods of treating resistant microorganisms has led to the creation of a combination of vaborbactam, which is a non-β-lactam β-lactamase inhibitor, with meropenem—an antibiotic from the carbapenem group [21,22].

### 2.1. Chemical Structure of Vaborbactam

Vaborbactam (VAB) (formerly RPX7009), with the molecular formula: 2-[(3R,6S)-2-hydroxy-3-[(2-thiophen-2-ylacetyl)amino]oxaborinan-6-yl]acetic acid (Figure 1), is a cyclic boronic acid pharmacophore inhibitor with serine protease activity (PubChem). The activity of the β-lactamase inhibitor is increased by the 2-thienoacytel group of vaborbactam. The compound has the ability to block enzymes of KPC (*K. pneumoniae* carabapenemase) bacterial strains and improves the activity of carbapenem antibiotics, especially in groups of CRE strains that carry genes encoding β-lactamases [23].

Vaborbactam is an inhibitor of type A and C β-lactamases [24]. The mechanism of its action is to enter the bacterial cell through porins OmpK35 and OmpK36. Therefore, the AcrAB-TolC multidrug efflux pump has a negligible effect on the action of vaborbactam [25].

### 2.2. Chemical Structure of Meropenem

Meropenem (MER), with the molecular formula (4R,5S,6S)-3-[[(3S,5S)-5-dimethylcarbamoylpyrrolidin-3-yl]thio]-6-[(1R)-1-hydroxyethyl]-4-acid methyl-7-oxo-1-azabicyclo[3,2,0]hept-2-ene-2-carboxylic acid (Figure 2), is a carbapenem that acts on the cell wall by inhibiting the cross-linking of its peptidoglycan, resulting in lysis and subsequent cell death. It is unstable in aqueous solution and has bactericidal activity against both Gram-positive and Gram-negative bacteria [26,27]. For maximum antibacterial potential, the %T > MIC (MIC—minimal inhibitory concentration; %T > MIC is the percentage of time that free-drug concentrations are higher than the MIC) of the free drug should be approximately 40%. In turn, for bacteriostatic activity, %T > MIC is approximately 20% [28,29]. The advantage of meropenem is that it is a weaker inducer of AmpC β-lactamases (cephalosporinases encoded on the chromosomes) produced by *Enterobacterales,* compared to other carbapenems, which prevents rapid development of resistance. It has high affinity for PBP 2, PBP3, and PBP4 (PBP; penicillin binding proteins). Moreover, it quickly penetrates the outer cell membrane of Gram-negative bacteria and has a highly stable hydrolytic effect [30]. Meropenem remains active against cephalosporinases and penicillinases, but loses stability against carbapenemases.

### 2.3. Combination of Meropenem with Vaborbactam

Meropenem/vaborbactam (MER-VAB) is a combination of two compounds under the trade name Vabomere^®^ (Menarini International Operations Luxembourg S.A., Luxembourg, Luxembourg). Vaborbactam reduces the MIC of meropenem by blocking the production of β-lactamases. A vaborbactam concentration of 8 µg/mL restores the activity of meropenem with an MIC value of ≤2 µg/mL for the most mutated bacterial strains [25]. MER-VAB was approved by the FDA (Food and Drug Agency) in 2017 for the treatment of patients over 18 years of age. It is registered for the treatment of urinary tract infections including pyelonephritis. Due to increasing multidrug resistance, the drug should only be used against sensitive bacteria [31].

### 2.4. Pharmacokinetic Properties

According to the study by Sabet et al., the half-life for a therapeutic dose (2 g MER/2 g VAB), administered in a 3 h infusion every 8 h, is 1.36 ± 0.07 and 1.47 ± 0.14 for MER and VAB, respectively. The maximum concentration (Cmax; mg/L) reaches the value of 34.47 ± 3.95 and 24.1 ± 5.84 for MER and VAB, respectively, whereas AUC (area under the ROC curve) 0–8 h/0–24 h (mg/L) is 132.63 ± 15.23/397.88 ± 45.70 and 105.57 ± 4.42/316.7 ± 13.26 for MER vs. VAB [32]. Both compounds have a similar half-life in the final process of drug elimination from the plasma [33]. It is important that meropenem/vaborbactam has a low degree of binding to plasma proteins, and administration in multiple doses does not cause accumulation in the body [34]. About 1/4 of the β-lactam ring of meropenem is metabolized in the liver. Hydrolysis metabolites and MER-VAB are excreted unchanged by the kidneys [35]. MER-VAB monotherapy shows a high degree of cure for CRE infections, as well as reduced nephrotoxicity in relation to the use of BAT (best available treatment) [36]. The use of MER-VAB affects glomerular filtration rate (GFR) by increasing the renal clearance value of 8.6–17.6 L/h, respectively, for VAB and 6.9–13.5 L/h for MER, where the range in healthy people is 5.4–7.2 L/h. This indicates active glomerular secretion and renal filtration [33]. It is worth noting that the pleural penetration is sufficient when the MER-VAB is used [37]. Taking into account the total drug concentration in plasma and based on the AUC0-8 ELF (epithelial lining fluid) values for meropenem, the intrapulmonary penetration is as much as 63%, while for vaborbactam it is 53% (3 doses of 2 g MER/2 g VAB in a 3 h prolonged infusion). The persistent ELF concentration above the MIC limit of 1 µg/mL and good penetration suggest the possibility of using MER-VAB therapy to treat HAP and ventilator-associated pneumonia [38].

### 2.5. Spectrum of Action

Meropenem is highly effective against microorganisms with ESBLs resistance. It is safe and very well tolerated, which contributes to the use of this antibiotic in the treatment of infections caused by Gram-negative bacteria [39]. Vaborbactam, on the other hand, has no antimicrobial effect, but blocks class A bacterial β-lactamases (TEM—plasmid-encoded β-lactamase in Gram-negative bacteria, CTX-M—preferential hydrolytic activity against cefotaxime, SHV—sulf-hydryl variable penicilinase) and C (FOX—AmpC β-lactamase gene, MIR—novel plasmid mediated β-lactamase, P99—β-lactamase from *Enterobacter cloacae* P99), as well as class A carbapenemase [25]. Data indicate that the addition of vaborbactam at a dose of 8 mg/L results in a reduction in the MIC by as much as 8–64 times. This dose of vaborbactam also reduces the MIC of meropenem to ≤8 mg/mL for all strains. Furthermore, combined with MER-VAB (MIC ≤ 1 μg/mL), it is active against *E. coli*, including the genes of the latest *bla*_KPC_ and mutations, obtaining significantly better results than in the case of meropenem alone (sensitivity 99.9%) with a decrease in MIC values. Compared to *K. pneumoniae*, the activity of MER-VAB increased from 88% to 98.8% at doses of 1 µg/mL MER and 4 µg/mL VAB. Additionally, in this case, MER-VAB lowers the MIC value and affects the KPC. A similar relationship has been observed in the species *Enterobacter cloacae* and *Enterobacter aerogenes*. However, for *A. baumannii*, *Stenotrophomonas maltophila*, and *P. aeruginosa,* the MER-VAB combination slightly changes the MIC value, which indicates the ineffectiveness of the therapy [40]. This was confirmed by in vitro studies, which proved the effect of MER-VAB on *Citrobacter freundii*, *Citrobacter koserii*, *E. cloacae*, *E. coli*, *E. aerogenes*, *K. pneumoniae*, *Klebsiella oxytoca*, *Morganella morganii*, *Proteus mirabilis*, *P. aeruginosa*, *Serratia marcescens*, and *Providencia* spp. Nevertheless, no effect has been shown on *A. baumannii* and *S. maltophilia* [31].

### 2.6. MER-VAB against Bacterial Resistance

Based on the available research, it can be concluded that MER-VAB is an alternative treatment for infections caused by bacteria belonging to the *Enterobacterales* producers of ESBLs, KPC, and AmpC, and, thus, effectively reduces the level of resistance. Moreover, it shows activity against MDR, XDR (extensively drug-resistant), and CRE microorganisms with MIC50 (MIC that inhibited 50% of tested microorganisms) and MIC90 (MIC that inhibited 90% of tested microorganisms) in comparison to other tested antibiotics [41].

MER-VAB is used in infections with MDR microorganisms that produce carbapenemases of types A (TEM-116, CTX-M, SHV, ESBLs) and C (FOX, MIR, P99), but is not used in the treatment of infections with strains producing carbapenemase types B and D. Therefore, it is not used in infections with bacteria showing MBL (metallo-β-lactamase)-type resistance (NDM—New Delhi metallo-β-lactamase, IMP—imipenemase, VIM—Verona-integron-imipenemase) and/or OXA-48 (oxacilinase-48)-type resistance, because the limited activity of this combination has been demonstrated against them [42,43]. However, the research of Biaga et al. clearly indicates that MER-VAB in combination with aztreonam and/or with ceftazidime and avibactam simultaneously (CAZ-AVI) may constitute an effective therapeutic option against aztreonam-resistant NDM strains and other serine-producing members of the *Enterobacterales* [44]. Moreover, the effect of MER-VAB on OXA-2 (oxacilinase-2), OXA-30 (oxacilinase-30), i.e., narrow-spectrum oxacillinases, as well as on carbapenemases of the KPC, FRI-1 (carbapenem-hydrolyzing Class A β-lactamase), and SME-2 (carbapenem-hydrolyzing β-lactamase Sme-2) type has been demonstrated [45]. A summary of the activity of the MER-VAB combination is shown in Table 1.

Interestingly, MER-VAB shows very high activity against carbapenem-resistant (CRE) *K. pneumoniae* strains with or without KPC-type resistance in the standard variant, or, additionally, in the CAZ-AVI combination. This may be an alternative to treatment for infections resistant to this combination, which is used as a first-line drug. The in vitro effects of MER-VAB against *Enterobacterales* KPC(+) are stronger than commonly used antibiotics, i.e., CAZ-AVI, tygecycline, and others [46]. According to research conducted by Schields et al., microorganisms belonging to the *Enterobacterales* (mainly *K. pneumoniae*) with CRE resistance isolated from critically ill patients (mainly with bacteremia and pneumonia) staying in the ICU were successfully eradicated in 65% of cases. This therapy prolonged survival in 90% of patients after 30 days of use [47].

The only critical point in using MER-VAB is that *K. pneumoniae* can withstand this combination due to mutations in the *bla*_KPC_ gene [48,49]. Since these genes are located in transposons, clonal expansion, i.e., the transmission of microorganisms with these genes, is the most dominant method of global KPC distribution.

Inactivation against bacteria with porin mutations related to excessive expression of efflux pumps has been demonstrated. Reduced activity occurs in microorganisms with inactivated OmpK35 and OmpK36, and overexpressing AcrA strains [49].

## 3. Clinical Application

### 3.1. Dosage

MER-VAB is administered at a dose of 4 g (2 g MER/2 g VAB) every 8 h as a 3 h prolonged infusion. The treatment period is up to 14 days [35]. Dosage for impaired renal function eGFR (mL/min/1.73 m^2^) is as follows: for eGFR ≥ 50, 2 g MER, 2 g VAB every 8 h; for eGFR 30–49, dosage has to be decreased to 1 g MER, 1 g VAB every 8 h; for eGFR 15–29, 1 g MER and 1 g VAB time has to be extended to 12 h; for eGFR < 15, dosage has to be decreased to 0.5 g MER and 0.5 g VAB every 12 h [50].

Due to the absence of data and safety evaluation, the dose for children has not been determined. Safety dosing research is ongoing [51]. The literature describes a case of the use of MER-VAB in a 4-year-old male child with short intestine syndrome, colostomy and gastrojejunal tube, bronchopulmonary dysplasia, and central line. Multiple central lines caused blood infections, which were treated with MER-VAB at a dose of 40 mg/kg every 6 h for 3 h as an infusion. The treatment lasted 14 days and resulted in clinical success, with *K. pneumoniae* KPC(+) being eliminated from the bloodstream [52].

### 3.2. Indications

According to the guidelines of the European Medicines Agency (EMA), MER-VAB is used for (i) complicated abdominal infections; (ii) hospital acquired pneumonia, including ventilator-associated pneumonia; (iii) blood infections; (iv) treatment of infections caused by aerobic Gram-negative microorganisms in patients with limited therapeutic options. Furthermore, according to FDA’s guidelines, MER-VAB is recommended for the therapy of complex urinary tract infections (cUTIs), including pyelonephritis kidney infection. Studies conducted by Shoulders et al. have demonstrated its efficacy in these types of infections [53]. The dose of meropenem and vaborbactam (2 g/2 g) every 8 h must be administered with uninterrupted renal function in each of the above cases, depending on the duration of the treatment [54].

### 3.3. Interactions

Administration of meropenem with valprionic acid reduces its concentration, which lowers the seizure threshold with the risk of an epileptic fit. Probenecid competes with meropenem for tubular secretion and increases the concentration of meropenem in the blood [35].

## 4. Antimicrobial Activity of Meropenem–Vaborbactam

MER-VAB’s antimicrobial activity in clinical environments was evaluated for the first time in phase 3 of the TANGO I (Targeting Antibiotic Non-susceptible Gram-Negative Organisms) multicenter, randomized, double-blind, double-dummy, active-control trial. The study was conducted with adults who were afflicted with complicated urinary tract infections (cUTI) and acute pyelonephritis (AP). Patients received meropenem/vaborbactam (2 g/2 g via 3 h infusion, every 8 h) or piperacilin/tazobactam (PIP-TAZ) (4 g/0.5 g via 30 min infusion every 8 h). Results showed that MER-VAB was noninferior to PIP-TAZ. Moreover overall success in the end of intravenous treatment indicated superiority of MER-VAB over PIP-TAZ in the range of clinical cure [55]. Primary end-points of TANGO I trial are shown in Table 2.

The incidence of side effects in both groups was comparable: 39.0% and 35.5%, respectively, in MER-VAB and PIP-TAZ. The most commonly isolated uropathogens in mMITT (microbiological modified-intent-to-treat) populations were *E. coli* (65.1% vs. 64.3%) and *K. pneumoniae* (15.6% vs. 15.4%). Other isolated pathogens (*P. mirabilis*, *E. cloacae*, and *P. aeruginosa)* were less numerous. Clinical cure efficiency in mMITT populations was 98.4% vs. 94.0% for *E. coli*, 100.0% vs. 100.0% for *P. mirabilis*, *E. cloacae*, and *P. aeruginosa*, and 96.7% vs. 100.0% for *K. pneumoniae* (respectively, for MER-VAB and PIP–TAZ), whereas microbiological eradication ratio was 71.2% vs. 58.1% *E. coli*, 63.3% vs. 50.0% *K. pneumoniae*, 50% vs. 75% *P. mirabilis*, 90% vs. 60% *E. cloacae*, and 100% vs. 30% *P. aeruginosa* (respectively, for MER-VAB and PIP-TAZ) [55].

MER-VAB’s clinical effectiveness was also evaluated in three phase randomized clinical trials of TANGO II. MER-VAB was compared to BAT (best available treatment—which was polytherapy of polymixin/aminoglycosides/carbapenems/tigecycline or monotherapy using CAZ-AVI) in the therapy of infections (cUTI, HABP/VABP—hospital-acquired-bacterial-pneumonia/ventilator-acquired-bacterial-pneumonia, AP, BSI—bloodstream infection, cIAI—complicated intra-abdominal infection) due to CRE [34]. The end-points of the TANGO II trial are shown in Table 3.

Patients were divided into two groups: those receiving MER-VAB therapy and those receiving BAT. Infections that were the most prevalent among patients in both groups were BSI, cUTI/AP, HABP/VABP, and cIAI. *K. pneumoniae* and *E. coli* were the most frequently isolated pathogens. CRE pathogens in both groups represented, respectively, 71.9% and 93.3% of the isolates. MER-VAB was more effective in overall clinical success in both EOT (end of treatment) and TOC (test of cure) in the mCRE-MITT population. Patients with confirmed cUTI/AP who received MER-VAB therapy scored a 75% result in overall clinical success vs. 50% outcome in BAT population. The clinical efficiency of administered therapy in TOC was, respectively, 33.3% vs. 50.0% (MER-VAB vs. BAT). Patients from the MER-VAB population also obtained a lower rate of day-28 mortality vs. BAT. The least mortality rate occurred among patients with HABP/VABP who received MER-VAB (22.2%) vs. BAT (44.4%). Treatment emergent adverse events (TEAE) were less frequent in the MER-VAB population (84%) vs. BAT (92%), as well as severe TEAE (14% vs. 28%) [36].

Shields et al. conducted a large-scale study on the clinical use of MER-VAB at the University of Pittsburgh Medical Center from 2017 to 2019. In this research, MER-VAB was used as a first-line treatment option against the CRE producing KPC. Clinical success was defined as a composite result with the following outcomes: (i) complete resolution of the symptoms of infection; (ii) absence of recurrence of infection; (iii) absence of microbiologic failure (isolating the same bacterial species for 7 days after MER-VAB treatment) within 30 days from the appearance of the infection. The study involved a cohort of 20 patients with several types of bacterial infections, i.e., BSI, bacterial pneumonia including VABP, tracheobronchitis, SSTI (skin and soft tissues infection), cIAI, and AP. The most commonly isolated pathogens were *K. pneumoniae* (n = 14), *Klebsiella oxytoca* (n = 2), and *E. coli* (n = 2). The monotherapy of MER-VAB was administered by i.v. infusion (2 g + 2 g every 8 h) in 16/20 cases. Only four patients additionally received another drug for >48 h (gentamicin inhalation (n = 2), gentamicin i.v. and inhalation (n = 1), and ciprofloxacin i.v. (n = 1)). The 30- and 90-day survival rates presented were, respectively, 90% and 80%. The overall clinical success was obtained in 13/20 patients (65%), whereas in the case of BSI and pneumonia, it amounted, respectively, to 63% and 67%. In the entire population, only one (1/20) case of severe TEAE was reported. Meanwhile, six patients developed the microbiologic failure (recurrent CRE infection (n = 3), persistent bacteremia, respiratory tract colonization, breakthrough infection during treatment). It is important to note that one of the recurrent bacterial isolates was resistant to MER-VAB. The overall clinical success and day-30 survival ratio achieved 65% and 90%, respectively, which are comparable to the TANGO II trial outcomes (59% and 84%) [46]. MER-VAB seems to be noninferior to CAZ-AVI, which obtained, respectively, 59% and 76% in the study of Shields et al. [56]. Finally, their study confirmed that TANGO II trial results were related to clinical efficiency of MER-VAB in the treatment of severe VABP. In the future, MER-VAB could be a possible alternative treatment for polymyxin in CRE pneumonia [47].

Tumbarello et al. described the clinical use of MER-VAB in a retrospective, observational study conducted in 12 Italian hospitals from 2020 to 2021. The study group consisted of 37 patients with confirmed *K. pneumoniae* producing KPC infection. Patients were hospitalized for BSI (23/37), LRTI (lower-respiratory-tract infection) (10/37), IAI (1/37), cUTI (2/37), and ABSSI (acute bacterial skin and skin structure infection) (1/37). MER-VAB (2 g + 2 g) was administered via 3 h i.v. infusion every 8 h for ≥72 h. Bacterial isolates were resistant to penicillins, extended spectrum cephalosporines, ciprofloxacin, and meropenem. In addition, 22 out of 37 patients’ isolates were confirmed to be resistant to CAZ-AVI. Clinical cure was achieved in 28/37 cases, while microbiologic eradication occurred in 25/37 cases. In-hospital mortality was observed for 24.3% of the patients. Deaths appeared only in patients with LRTI (n = 3) and BSI (n = 6) and affected mainly sufferers above 60 years old. The recurrent in-hospital infection was observed in three cases but was successfully treated using MER-VAB with colistin or fosfomycin combined therapy. Tumbarello’s study revealed that MER-VAB is very efficient in the treatment of CAZ-AVI—resistant, KPC(+) *K. pneumoniae* infections (CAZ-AVI—resistant pathogens represented at least 60% of all isolates). This is a significant condition in the era of increasing microbial resistance to CAZ-AVI [57].

Outcomes of Carvalhaes et al.’s analysis about in vitro susceptibility of bacterial isolates collected from patients cured in USA hospitals in 2014–2018 revealed 99.9% antimicrobial efficiency of MER-VAB against *Enterobacterales* (4790 isolates) and 85.9% against *P. aeruginosa* (3193 isolates). Isolates were collected from patients with confirmed HABP/VABP infections. MER-VAB’s in vitro activity against CRE pathogens (resistant to many other antimicrobial agents) oscillated above 99%. MER-VAB’s ability to treat CRE etiology HABP/VABP is noteworthy as it also has the potential to treat MDR/XDR *P. aeruginosa* isolates [58].

Castanheira et al., in a similar study involving CRE isolates (with a predominance of *K. pneumoniae* 77/152 and *E. cloacae* 27/152) collected in USA hospitals in 2016–2018, revealed that MER-VAB inhibited growth of 95.4% isolates, whereas β-lactam comparators only inhibited 6.6%. Comparable results to MER-VAB in the inhibition of growth of CRE were achieved only by tigecycline (96.7%). The study confirmed that MER-VAB is more efficient than combination therapy and has greater effectiveness than comparators commonly used in the treatment of CRE etiology infections [59].

Alosaimy et al. carried out a multicenter, real-world analysis of the clinical outcomes of MER-VAB therapy for GNB (Gram-negative bacteria). Patients who were qualified in the sample population (n = 40) reported serious bacterial infections. The most commonly confirmed types of infections were pneumonia (32.5%) and UTI (20%) caused particularly by *Enterobacterales* (*K. pneumoniae* 46.7%, *E. cloacae* 20.0%, and *E. coli* 13.3%). Clinical success was achieved in 70% of all cases, and in-hospital mortality rate was 7.5% (3/40). In all other cases, recurrent infection or presence of sign or symptoms of infection defined clinical failure. In post hoc analysis, day-60 and day-90 mortality rates reached, respectively, 15.0% and 22.5%. In terms of clinical outcomes and safety of MER-VAB, the analysis complements the results of the TANGO II trial [60].

Another multicenter, retrospective analysis of clinical efficiency of MER-VAB was conducted in USA in 2017–2020. Patients who qualified for the study (n = 126) were infected by MDR GNB, including CRE pathogens (mainly *K. pneumoniae*, *E. coli*, and *Enterobacter* spp.). The overall number of patients with confirmed CRE infection was 99. In addition, patients (n = 10) with confirmed non-CRE infections (*A. baumannii* and *P. aeruginosa*) were included in the study population and also received MER-VAB therapy. Infections in hospitals accounted for 58.7% of all cases. The most prevalent infections were respiratory tract (38.1%), abdominal cavity (19.0%), and urinary tract (13.5%). The mortality rate on day 30 was 18.3% (23/126), and 15 cases (11.9%) of repeated infection were reported. On the other hand, the mortality rate on day 90 eventually reached 31.7%. The positive clinical efficacy of MER-VAB against *P. aeruginosa* (day-90 mortality rate 1/8, recurrence of infection within 30 days to 2/8) is encouraging; however, the population of the study was not sufficiently large to draw some clear conclusions [61].

Comparisons of the clinical efficacy of CAZ-AVI and MER-VAB for CRE infections were carried out in a retrospective multicenter study between 2015 and 2018. The research included 131 patients in two cohorts receiving CAZ-AVI or MER-VAB (respectively, n = 105 and n = 26 patients) *K. pneumoniae* and *E. cloacae* were the most frequently isolated pathogens in both groups. The primary end-point of the study was overall clinical success (defined as a composite result of microbiologic eradication and resolution of the signs and symptoms of infection) with scores of, respectively, 62% vs. 69%. Meanwhile, the secondary end-points of trial (day-30, day-90 mortality ratio) revealed no crucial fluctuation, much like the issue of adverse events (36/105 vs. 6/26). Recurrent infection was confirmed in 14.3% of patients in the CAZ-AVI cohort and in 11.5% in the MER-VAB cohort. Post hoc analysis showed that bacterial drug resistance developed in patients with recurrent infections receiving CAZ-AVI monotherapy (n = 3). The treatment of MER-VAB and polytherapy of CAZ-AVI with other drugs did not cause this problem. In addition, CAZ-AVI monotherapy was associated with a greater need to switch to combined therapy (64/105 vs. 4/26). Moreover, the rate of drug resistance development among pathogens in CAZ-AVI monotherapy was higher than in MER-VAB treatment. Overall, MER-VAB clinical success was similar to the results of the TANGO II trial (respectively 69% vs. 64%) [62].

Interesting conclusions were provided by the study of the in vitro activity of vaborbactam combined with β-lactam antibiotics against clinical isolates of the MDR *Mycobacterium abscessus complex* (MABC). The antimicrobial activity of carbapenems and certain cephalosporines (particularly cefuroxime-II generation cephalosporine) against MABC was increased by vaborbactam. Vaborbactam, like relebactam and avibactam, is more efficient than classical inhibitors (e.g., sulbactam) against MB (*Mycobacterium* spp.) because it does not have a β-lactam ring in its structure. In the future, MER-VAB may be considered as a treatment strategy against MABC because of the restoring susceptibility to, for example, meropenem and cefuroxime among mycobacteria [63].

Gainey et al. described antimicrobial therapy implemented in the case of a pediatric patient suffering from cystic fibrosis (CF) complicated with MDR *Achromobacter* spp. etiology pneumonia who was successfully treated via a combination of MER-VAB, cefiderocol, and phage therapy. The therapy was well tolerated by the patient. An in vitro study by Caverly et al. revealed that MER-VAB is an effective method for managing CF complicated with *Achromobacter* spp., *Burkholderia gladioli,* and *Burkholderia cepacia* (including MDR/XDR strains) infection. MER-VAB had low antimicrobial activity (like other tested BL + BLI (β-lactam + β-lactamase inhibitor)) against *Pandoraea* spp. and *S. maltophilia* [64,65].

Antimicrobial activity of MER-VAB against clinical isolates of *Enterobacterales* and *P. aeruginosa* was tested in the study of Dee Shortridge et al. Pathogens were collected in 21 European countries in the years 2014–2019. The total amount of isolates was 6846 (*K. pneumoniae*, n = 1877; *E. coli,* n = 1646*; P. aeruginosa*, n = 3567) [66]. Bacterial strains were collected mainly from patients hospitalized for HABP/VABP. Total microbial susceptibility to MER-VAB reached 98% within *Enterobacterales* and 82.1% in *P. aeruginosa*. Authors reported that sensitivity to MER-VAB was higher in strains from Western Europe than Eastern Europe. The study result could possibly be explained by the fact that OXA-48 resistance is more prevalent than KPC in Eastern Europe. MER-VAB inhibited the growth of 99.1% KPC(+) isolates vs. 40.5% OXA-48(+) isolates. Moreover MER-VAB obtained the highest efficiency (98%) among comparators used in the study against *Enterobacterales* isolates. Furthermore, MER-VAB was also the most active β-lactam antibiotic against *P. aeruginosa* (82.1%). The higher result reached only two comparators: colistin and amikacin [66].

In vitro antimicrobial activity of MER-VAB and other BL + BLI combinations (CAZ-AVI, I-R-C (imipenem-relebactam-cilastatine), C-TZB (ceftolozane/tazobactam)) against *P. aeruginosa* strains isolated from Western (WE) and Eastern Europe (EE) patients was evaluated in the study of Sader et al. Isolates were collected from patients with confirmed bacterial pneumonia. All of the BL+BLI revealed antimicrobial activity against >90% WE isolates, and MER-VAB showed the lowest efficiency 91% (MIC50 vs. 90 = 0.5 vs. 8 mg/L) versus comparators (CAZ-AVI = 97.2%; I-R-C = 94.5%; C-TZB = 94.3%). MER-VAB was also the least effective (83.5%) vs. comparators against EE strains. Moreover, the case revealed that MER-VAB had also the lowest antimicrobial efficiency (among BL + BLI comparators) against a *P. aeruginosa* strain resistant to PIP–TAZ, CAZ, MER, and IMI (imipenem) [67].

A clinical study by Belati et al. revealed that the combination of MER-VAB and aztreonam could be used as a treatment option for KPC(+), CAZ-AVI-resistant *K. pneumoniae*. All isolates were collected from the patients with confirmed severe BSI. Other authors described the case of MER-VAB as a salvage therapy against KPC(+), *K. pneumoniae* resistant to CAZ-AVI, and cefiderocol. Results showed that complete resolution and absence of recurrent infection were observed in a previously critically ill patient [68,69].

Sabet et al. described the potential clinical use of MER-VAB against *A. baumannii* and *P. aeruginosa*. In the study, authors proved in vivo efficiency of MER-VAB and meropenem therapy using a murine model of GNB infection complicated with neutropenia. MER-VAB achieved a higher eradication rate than meropenem alone against *P. aeruginosa* isolates. MER-VAB obtained excellent antimicrobial activity in *P. aeruginosa* and *A. baumannii* etiology infections. The results of this study correlate with similar articles, e.g., Sabet et al. and Weiss et al. (murine model of pyelonephritis) [70,71,72].

Bovo et al. conducted a study between 2018 and 2020 on the in vitro sensitivity of clinical bacterial isolates (collected from patients with bacteremia) to MER-VAB and other novel BL + BLI. In total, 133 bacterial strains of *K. pneumoniae* (within 87.2% strains were KPC(+)) were evaluated in this study. During the years 2018–2020, the in vitro antimicrobial activity of MER-VAB fluctuated from 85% to 97% [73].

Tan X et al. provided a comprehensive overview of the current knowledge on the treatment of cUTI with carbapenemes or their combinations with BLI. The authors revealed that MER-VAB, ertapenem, and biapenem are characterized by higher overall clinical success rate in this type of infection. MER-VAB and ertapenem also achieved a higher microbiologic eradication rate than other comparators. On the other hand, MER-VAB, similarly to imipenem/cilastatin, obtained a higher percentage of adverse events. The authors postulated that MER-VAB should be used as a first-choice drug in cUTIs of CRE etiology [74]. An overview of the antimicrobial activity of MER-VAB is summarized in Table 4.

## 5. Adverse Events

Adverse events (Aes) related to MER-VAB therapy were described in clinical trials of phase 3: TANGO I and TANGO II. In the first case (study population n = 272), AE occurred in 39% of patients. The most common AE was headache (8.8%), and other AEs were diarrhea (3.3%), nausea (1.8%), asymptomatic bacteriuria (1.5%), catheter site phlebitis (1.8%), infusion site phlebitis (2.2%), UTI (1.5%), hypokalemia (1.1%), vaginal infection (0.4%), alanine aminotransferase increased (1.8%), aspartate aminotransferase increased (1.5%), anemia (0.7%), and pyrexia (1.5%). MER-VAB was well tolerated by patients, and only 2.6% of them had to discontinue therapy because of severe TEAE [54]. In the TANGO second trial, TEAEs were reported in 84% (42/50) patients. The most commonly represented AEs were diarrhea (12%), anemia (10%), hypokalemia (10%), hypotension (8%), sepsis (4%), septic shock (2%), and AKI (2%). MER-VAB also showed a lower nephrotoxicity rate than BAT [36].

Similar amounts of AEs were described in a phase 1 study of the safety, tolerability, and pharmacokinetics of MER-VAB. TEAEs occurred in 62/80 patients (77.5%); 26 of them (study population n = 37) were connected with MER-VAB administration and 6 of them (study population n = 8) with administration vaborbactam alone. The most commonly observed AEs were infusion site phlebitis and infusion site pain [33].

Shields et al. indicated a single case of severe TEAE that occurred in the study population (n = 20). One patient developed severe eosinophilia after 19 days of MER-VAB i.v. therapy [47]. Tumbarello et al. recorded a single case of severe TEAE (1/37) in the form of leukopenia and thrombocytopenia [57].

Alosaimy et al. revealed that MER-VAB therapy involved low risk of nephrotoxicity (AKI) and diarrhea or *Clostridioides difficile* infectious complications. However, one of the patients developed a severe dermatological reaction—Stevens–Johnson syndrome/toxic epidermal necrolysis (SJS/TEN)—which was a fatal complication of the therapy [60]. Moreover, the authors proved that MER-VAB treatment had low risk of hepatotoxicity (0.8%). A total of 3.1% of patients experienced AEs, which is a lower number in comparison to the TANGO I and TANGO II trials [61].

Ackley and co-authors analyzed the safety of MER-VAB and CAZ-AVI treatment. In their study, MER-VAB showed lower rate of AEs than CAZ-AVI, respectively, 23.1% (6/26) vs. 34.3% (36/105). Nephrotoxicity (defined as AKI) was the most commonly observed form of AE (14.3%). Other AEs observed in the study were leukopenia (7.7%) and rash (3.9%) [62].

## 6. Conclusions

The emergence of multidrug-resistant strains represents one of the greatest dangers to health and life around the world. The antibiotic resistance rate is increasing rapidly, leading to the shortage of therapeutic options. Therefore, antibiotic use in medicine must be rationalized and the development of new compounds for future drug should be increased. A promising trend is the preparation of a method to deactivate the resistance mechanism due to the actions of antibiotic inhibitors. In this review, we took a closer look at the most recent combination of meropenem, a β-lactam antibiotic, with vaborbactam as an inhibitor (MER-VAB). In the time of antimicrobial-drug-resistance growth, novel combinations of BL+BLI may become important treatment options against Gram-negative bacteria. MER-VAB is predicted by many authors to play a greater role in the treatment of critically ill patients, particularly those with HABP and VABP. Many of them postulate using MER-VAB as a first-choice treatment option for CRE-related pneumonia (including hospital and ventilator-acquired pneumonia). The good tolerability and clinical efficiency confirmed in the TANGO I trial presents MER-VAB as an interesting antimicrobial therapy option in cUTIs caused by CRE pathogens. Moreover, there are suggestions for the use of MER-VAB in cUTIs as a treatment for CRE KPC(+), with AP included. Based on the data provided in the article, it is possible to predict the trends in the growing role of novel BL + BLI medicines. In the near future, it will be noted that combinations of drugs such as MER-VAB, CAZ-AVI, and cefiderocol will become the first-line antimicrobials against the spread of CRE pathogens, especially in patients in ICUs.

## Figures and Tables

**Figure 1 antibiotics-12-01612-f001:**
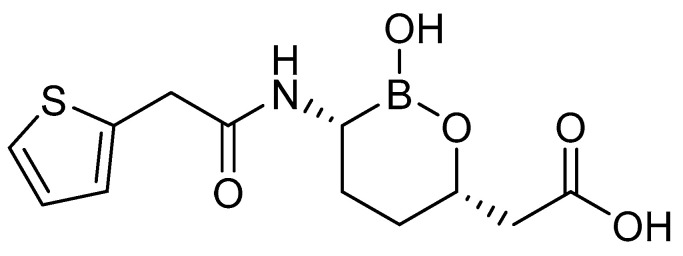
The 2D (two-dimensional) chemical structure of vaborbactam.

**Figure 2 antibiotics-12-01612-f002:**
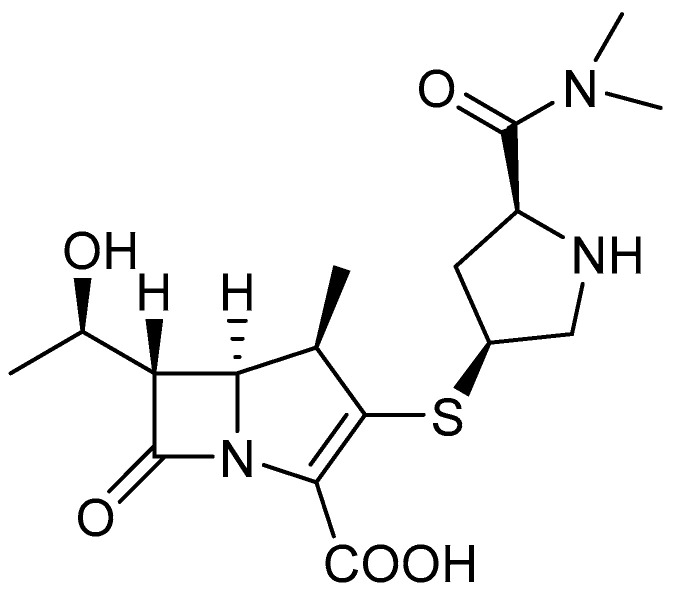
The 2D (two-dimensional) chemical structure of meropenem.

**Table 1 antibiotics-12-01612-t001:** Scheme of MER-VAB action on individual types of resistance in microorganisms.

Class According to Ambler	Effect of the Vaborbactam	β-Lactamases
Class A	Inhibition	Serine carbapenemases: KPC, NMC-A, SME-2
		ESBL: SHV-2, PER-1
		Narrow spectrum: TEM-1, TEM-2
Class B	Lack of inhibition	Metalo-β-lactamases: VIM, NDM-1
Class C	Inhibition	Cephalosporinases: AmpC, P99, ACT-1
Class D	Lack of inhibition	OXA-48

Legend: KPC—*K. pneumoniae* carabapenemase; NMC-A—chromosomal-encoded class A β-lactamases first isolated from *Enterobacter cloacae* (clinical strain NOR-1); SME-2—carbapenem-hydrolyzing β-lactamase Sme-2; ESBL—extended-spectrum β-lactamases; SHV-2—sulf-hydryl variable penicillinase; PER-1—Pseudomonasextended resistant β-lactamase; TEM-1—plasmid-encoded β-lactamase in Gram-negative bacteria; TEM-2—plasmid-encoded β-lactamase in Gram-negative bacteria; VIM—Verona-integron-imipenemase; NDM-1—New Delhi metallo-β-lactamase-1; AmpC—cephalosporinases encoded on the chromosomes of many of the *Enterobacteriaceae*; P99—β-lactamase from *Enterobacter cloacae* P99; ACT-1—plasmid-encoded AmpC β-lactamase; OXA-48—oxacilinase-48.

**Table 2 antibiotics-12-01612-t002:** TANGO I trial primary end-points.

Primary End-Points	Meropenem/ Vaborbactam	Piperacilin/ Tazobactam
Overall success defined as clinical cure or improvement and microbial eradication composite * (*p* < 0.001)	98.4%	94.0%
Microbiological eradication in mMITT (*p* < 0.001)	66.7%	57.7%
Microbiological eradication in ME (*p* < 0.001)	66.3%	60.4%

* Clinical cure defined as complete subsidence or significant improvement of initial signs and symptoms of cUTIs or AP and microbial eradication defined as baseline pathogens reduced to <10^4^ CFU/mL. Primary end-point by FDA. Trial was performed among population of patients, divided into two groups: research sample n = 274 (MER-VAB therapy) and control n = 276 (PIP–TAZ therapy); mMITT—microbiological modified-intent-to-treat population, which included all the patients who presented baseline pathogens of ≥10^5^ CFU/mL in urine culture or the same bacterial pathogen present in concurrent blood and urine cultures. Primary end-point by EMA (European Medicines Agency); ME—microbiological evaluable analysis.

**Table 3 antibiotics-12-01612-t003:** End-points of the TANGO II trial.

Primary End-Points of TANGO II Trial	Meropenem/ Vaborbactam	BAT
Overall clinical success * in EOT (*p* = 0.03)	65.6%	33.3%
Microbiologic cure in EOT	65.6%	40.0%
Overall clinical success in TOC	59.4%	26.7%
Microbiologic cure in TOC	53.1%	33.3%
Day-28 mortality in mCRE-MITT population	15.6%	33.3%

* Overall clinical success was defined as a composite result of clinical and microbiological cure at EOT and TOC. Clinical cure (in accordance with FDA) was defined as a total resolution of baseline signs and symptoms of infection without need of further antimicrobial treatment or surgery in case of cIAI; microbiologic cure was defined as microbiological eradication or presumed eradication (clinical cure in absence of sample for repeat culture) at EOT and TOC; mCRE-MITT (microbiologic-CRE-modified-intent-to-treat population)—population of patients (with microbiologically confirmed CRE-pathogen related infection) who received ≥1 dose of study drug. MER-VAB: n = 32, BAT: n = 15; EOT—end of treatment; TOC—test of cure (7 ± 2 after EOT).

**Table 4 antibiotics-12-01612-t004:** Antimicrobial activity of meropenem–vaborbactam.

Microbiological Characteristics	No. of Patients	Infection Type	No. of CRE Pathogens	Median Duration of Treatment	Clinical Cure	Therapy Scheme	In-Hospital Mortality	Severe TEAE	References
*K. pneumoniae* (14/20), *K. oxytoca* (2/20), *E. coli* (2/20), *E. cloacae* (1/20), *C. freundii* (1/20)	n = 20	- BSI (8/20), - Pneumonia (6/20 - incl. 5/6 HABP) - Tracheobronchitis - (2/20 incl. 1VATB) - SSTI (2/20) - AP (1/20) - cIAI (1/20)	18/20 (90%)	8 days	65% (13/20)	MER-VAB MT in 80% (16/20); PT > 48 h (4/20) (GM inh. (n = 2), GM i.v. and inh. (n = 1), CIP i.v. (n = 1))	2/20	1/20 eosinophilia	[46]
*K. pneumoniae* (100%)	n = 37	- BSI (23/37), - LRTI (10/37), - IAI (1/37), - cUTI (2/37), - SSTI (1/37)	100%	13.5 days	75.6% (28/37)	MER-VAB MT (14/37); PT ((+1 drug), n = 17: FOS (6), COL (6), TGC (3), GM (1), AKN (1); (+ ≥2 drugs) n = 6)	9/37	1/37 leukopenia, thrombocytopenia	[56]
*Enterobacterales* (86.7%) 39/45: *K. pneumoniae* (46.7%) *E. cloacae* (20%) *E. coli* (13.3%) *B. cepacia* (6.6%) *A. baumannii* (2.2%) *P. aeruginosa* (4.4%)	n = 40	- Nosocomial infections 18/40 - cUTI 8/40 - IAI 5/40 - SSTI 5/40 - BSI 2/40	33/39 (84.6%)	12 days	70% (28/40)	MER-VAB MT; PT ((+ 1 drug): MNO/LEV/AKN 37.5% (12/40); 8/40 TOB/COL; 4/40 changed after 72 h MER-VAB for MNO/CAZ-AVI)	3/40	1/40 SJS/TEN	[59]
*K. pneumoniae* (42.1%) *E. coli* (19.8%) *E. cloacae* (16.7%) *P. aeruginosa* (8.7%) other (12.7%)	n = 129	- Pneumonia (38.1% incl. VABP 19.8%) - IAI (19.0%) - UTI (13.5%) - SSTI (10.3%) - BSI (9.5%) - Other (9.6%)	78.6%	11.7 days	ND	MER-VAB MT; PT (34.1% + ≥1 drug for ≥48 h)	30/126	4/126 SJS/TEN; hepatotoxiticy, AKI	[60]
*K. pneumoniae*, *E. coli*, *Enterobacter* spp., *Citrobacter* spp., *Serratia* spp.	n = 131 (incl. n = 105 CAZ-AVI n = 26 MER-VAB)	- MER-VAB cohort: - Primary BSI 1/26 - Secondary bacteriemia 8/26 - Nosocomial infections 10/26 - IAI 5/26 - SSTI 2/26	ND	MER-VAB cohort: 12.3 days CAZ-AVI cohort: 10.8 days	MER-VAB cohort: 69.2% CAZ-AVI cohort: 61.9%	MER-VAB cohort: MER-VAB MT; PT (4/26) (+ ≥1 drug) PXB/COL/TGC	MER-VAB cohort: 3/26	6/26 (3/6 AKI; 2/6 leukopenia;1/6 SJS)	[61]

Antimicrobial agents: GM (gentamycin), AKN (amikacin), TOB (tobramycin), TGC (tygecycline), COL (colistin), PXB (polymixin B), MNO (minocyclin), LEV (levofloxacin), FOS (fosfomycin), CIP (ciprofloxacin). Drug administration: i.v.—intravenous; inh.—inhalation. Severe TEAE: SJS/TEN—Stevens–Johnson syndrome/toxic epidermal necrolysis; AKI—acute kidney injury; infection type: VATB—ventilatory acquired tracheobronchitis. Abbreviations: ND—no data; MT—monotherapy; PT—polytherapy; incl.—include.

## Data Availability

Not applicable.

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
