# Peer review of "Meropenem/Vaborbactam: β-Lactam/β-Lactamase Inhibitor Combination, the Future in Eradicating Multidrug Resistance"

_antibiotics, 2023, doi:10.3390/antibiotics12111612_

Round 1
Reviewer 1 Report
Comments and Suggestions for Authors
The manuscript by Anna Duda-Madej reviewed the effect of meropenem in combination with a vaborbactam as potential antibiotic treatment. In general, the quality of the manuscript has not met the publication standards, I would like to reject the manuscript in current form.
1. There are lot of format errors, for example, line 398: n=40 should be n = 40. Line 392: 96,7% should be 96.7%.
2. The chem drawing of figure 1 and 2 is very sketchy.
3. There are a lot of abbreviations that need to be explained, which make the manuscript hard to understand. For example, Lin2 215 “Moreover, it shows activity against MDR, XDR, and CRE microorganisms with MIC50 215 and MIC90 lower than for other antibiotics in the comparative set.”
4. The manuscript just reiterates the abstracts of the cited papers, there is no novelty for the community. The author should summarize and have their own ideas or opinions.
Comments on the Quality of English LanguageThe manuscript needs extensive editing and careful proofreading.
Author Response
Dear Reviewer
Thank you very much for the time you took to read our Manuscript. We perfectly understand your negative decision. However, we would like to point out that we have revised our Manuscript in accordance with the recommendations of the other two Reviewers. In our proceedings, we also took into account these comments we received from you.
1 There are a lot of format errors, for example, line 398: n=40 should be n = 40. Line 392: 96.7% should be 96.7%.
Thank you for pointing this out. We have corrected all such errors and similar ones throughout the Manuscript.
- the chem drawing of figure 1 and 2 is very sketchy.
Thank you. I agree, you are right that the quality of figure 1 and 2 was poor. Both patterns were drawn in 2D and 3D in a special program.
3 There are a lot of abbreviations that need to be explained, which make the manuscript hard to understand. For example, Lin2 215 "Moreover, it shows activity against MDR, XDR, and CRE microorganisms with MIC50 215 and MIC90 lower than for other antibiotics in the comparative set."
Thank you very much for your valuable comment. Any abbreviations that may have made the text difficult to understand have been expanded.
- the manuscript just reiterates the abstracts of the cited papers, there is no novelty for the community. The author should summarize and have their own ideas or opinions.
Our aim in this review was to collect information on the efficacy of therapy using meropenem in combination with vaborbactam. We wanted to reach an audience such as doctors to increase their knowledge of the management of infections with the etiology of resistant G-negative strains (e.g., CPE)
As recommended, the work was read and revised by a native speaker.
We will understand if you stay with your decision. However, we hope that the corrections we have made have improved the quality of the Manuscript from your point of view.
Best regards,
Anna Duda-Madej
Reviewer 2 Report
Comments and Suggestions for Authors
The authors have made a good attempt to review the potential of the combination of Meropenem and Vaborbactam in addressing multiresistance. However, some issues need to be addressed or improved.
1. Line 110: “In this review, we have compiled all the data on the combination of a beta-lactam antibiotic with an inhibitor, meropenem with a vaborbactam.”
No justification for this claim. How were these data sourced to confirm that all existing information have been captured? Any specific search strategy? Also, there was no mention of how grey literatures and other unpublished data were accessed. If otherwise, it should be specified that a compilation of published data was done.
2. The expressions in the entire manuscript should be carefully checked. There are a couple of punctuations, spellings, and grammatical errors that affect the reading. A few areas are highlighted below:
- Line 59: highly mortality
- Line 71: “By nature following from the structure of the cell wall, it is Gram-negative bacteria have higher resistance than Gram-positive bacteria.”
- Line 154: penicilliases.
- Line 160: “…name Vabomere ® Vaborbactam reduces…”
- Line 166: “…every 8his 1.36 ± 0.07 and…”
- Line 197: “all strains What’s more…”
- Line 275: “…every 8hin the form…”
- Line 285: “…among population of adults suffered from complicated…”
- Line 347: “…20 patients with confirmed: BSI, bacterial…”
- Line 358: “Overall clinical success and 30-Day survival ratio finally obtained 65% and 90% what make them similar to TANGO II trial outcomes (59% and 84% respectively).”
- Line 386: “Importantly MER-VAB was also active against MDR/XDR P. aeruginosa isolates what makes him interesting treatment solution in CRE HABP/VABP.”
- Line 449: “…reports form Caverly et al. study.”
- Line 529: “AE occurred in 3,1% of patients what gives a lower result than in…”
- Line 547: “…in treatment critically ill patients suffered from…”
Comments on the Quality of English LanguageEnglish language needs to be improved in this manuscript.
Author Response
Dear Reviewer
Thank you very much for the time you took to read our Manuscript. Thank you for your valuable comments, which we have taken into account as follows:
1.Line 110: “In this review, we have compiled all the data on the combination of a beta-lactam antibiotic with an inhibitor, meropenem with a vaborbactam.”
Thank you for pointing out our unfortunate statement. It has been replaced as follows:
“ In this review, we have compiled the data on the combination of a β-lactam antibiotic with an inhibitor, meropenem with a vaborbactam.”
-Line 59: highly mortality
Thank you for pointing out our error. We threw out the word "highly"
-Line 71: “By nature following from the structure of the cell wall, it is Gram-negative bacteria have higher resistance than Gram-positive bacteria.”
We have changed that sentence. It currently reads: “Gram-negative bacteria show higher resistance than Gram-positive bacteria. They have a various abilities to learn new ways of failing at therapy and are easily transferring their genetic material (including antibiotic resistance genes) to other bacteria.”
-Line 154: penicilliases.
Thank you for pointing out the error. We have corrected it.
-Line 160: “…name Vabomere ® Vaborbactam reduces…”
Thank you for pointing out the missing period. It has been inserted.
-Line 166: “…every 8his 1.36 ± 0.07 and…”
Thank you for pointing out the missing space and period. They have been inserted.
According to the study by Sabet et al. , the half-life for a therapeutic dose (2g MER/2g VAB), administered in a 3-hour infusion every 8h, is 1.36 ± 0.07 and 1.47 ± 0.14 for MER and VAB, respectively.
-Line 197: “all strains What’s more…”
Thank you for pointing out the missing space. It has been inserted.
-Line 275: “…every 8hin the form…”
A space has been added in the sentence.
-Line 285: “…among population of adults suffered from complicated…”
The sentence with the not very accurate wording has been replaced with the following:
The study was conducted with adults who were afflicted with complicated urinary tract infections (cUTI) and acute pyelonephritis (AP).
-Line 347: “…20 patients with confirmed: BSI, bacterial…”
The missing items were added to the sentence. It now looks as follows:
„Study involved a cohort of 20 patients with several types of bacterial infections, i.e., BSI, bacterial pneumonia including VABP, tracheobronchitis, SSTI (Skin and Soft Tissuses Infection), cIAI and AP”.
-Line 358: “Overall clinical success and 30-Day survival ratio finally obtained 65% and 90% what make them similar to TANGO II trial outcomes (59% and 84% respectively).”
A sentence that is not very clear has been replaced with another one more clear in our opinion:
„The overall clinical success and Day-30 survival ratio achieved 65% and 90%, respectively, which are comparable to the TANGO II trial outcomes (59% and 84%)”.
-Line 386: “Importantly MER-VAB was also active against MDR/XDR P. aeruginosa isolates what makes him interesting treatment solution in CRE HABP/VABP.”
The earlier sentence was replaced with the following:
„MER-VAB's ability to treat CRE etiology HABP/VABP is noteworthy as it also has the potential to treat MDR/XDR P. aeruginosa isolates”.
-Line 449: “…reports form Caverly et al. study.”
The sentence was replaced with the following:
„An in vitro study by Caverly et al. revealed that MER-VAB is an effective method for managing CF complicated with Achromobacter spp, Burkholderia gladioli and Burkholderia cepacia (including MDR/XDR strains) infection”.
-Line 529: “AE occurred in 3,1% of patients what gives a lower result than in…”
Missing abbreviation explanations have been added to the Manuscript. In % expressions "," was changed to ".".
„The 3.1% of patients experienced AEs, which is a lower number in comparison to the TANGO I and TANGO II trials”.
-Line 547: “…in treatment critically ill patients suffered from…”
According to the suggestion, the sentence was changed to the following:
MER-VAB is predicted by many authors to play a greater role in the treatment of critically ill patients, particularly those with HABP and VABP
As recommended, the work was read and revised by a native speaker.
I hope that the corrections we have made are satisfactory and improve the quality of the Manuscript.
Best regards,
Anna Duda-Madej
Reviewer 3 Report
Comments and Suggestions for Authors
Comments on the Quality of English LanguageEnglish needs minor changes.
Author Response
Dear Reviewer
Thank you very much for the time you took to read our Manuscript.
- The title needs to be changed.
My recommendation is “Meropenem/vaborbactam: β-lactam/β-lactamase inhibitor
combination, the future in eradicating multidrug resistance”
Thank you for the suggestion to change the topic. The title has been changed as recommended.
- Keywords should be revised. “β-lactam/β-lactamase inhibitor” should be added instead of “cefiderocol” and “ceftazidime/avibactam.”
Thank you for pointing out the unnecessary keywords and replacing them with ones more related to Manuscript. The corrections have been made. And now the Manuscript contains the keywords: β-lactam/β-lactamase inhibitor; carbapenem-resistant Enterobacterales (CRE); Klebsiella pneumoniae carbapenemase (KPC); meropenem; multidrug resistance; vaborbactam
- All abbreviations should be reviewed throughout the article. Some abbreviations do not
have any extensions and prevent the review from being understood. All abbreviations
should be defined in the first place they appear.
Thank you for pointing it out. In fact, we missed some of the abbreviations. Now all abbreviations are explained in detail where they are first used. This will certainly contribute to a better understanding of the contents of the Manuscript.
- The introduction appears disorganized, as it attempts to describe the same concepts using
different words. There's a need to merge and revise certain paragraphs. In the introduction,
it's important to highlight the properties and significance of beta-lactams and beta-
lactamases, with a specific focus on carbapenems. Additionally, the introduction could
include mechanisms of action of beta-lactams and beta-lactamases.
The introduction has been rearranged. Some paragraphs were linked together, others were rewritten, and others were swapped places. In addition, information on β-lactam antibiotics (mainly carbapenems) has been added, including mechanisms of resistance to them.
„The cornerstone of antibiotic therapy is the largest and safest group which refers to β-lactam antibiotics. They inactivate transpeptidases, carboxypeptidases and endopeptidases involved in the synthesis of the bacterial cell wall. Carbapenems, which belong to this group of antibiotics, represent a last resort in the therapy of infections caused by Gram-negative Enterobacteriaceae [18]. However, since the β-lactam ring structure is sensitive to the action of bacterial enzymes, β-lactamases, which degrade and inactivate the antibiotic, inhibitors of these enzymes are a key factors. So far, several of them have been recognized, i.e., clavulanic acid, sulbactam, tazobactam, avibactam. However, in many cases, they have already lost their effectiveness, as bacteria have adapted to the new conditions by producing newer and newer enzymes (including carbapenemases)”.
- Line 147. Explain what is “T%>MIC”.
T%>MIC is the percentage of time that free-drug concentrations are higher than the MIC. This explanation has been added to the text.
„T%>MIC (MIC - minimal inhibitory concetration; T%>MIC is the percentage of time that free-drug concentrations are higher than the MIC)”.
- Line 158, Section 2.3. Include the drug's history in this section. When was it FDA-approved, and for which diseases?
Thank you for your attention. Information has been added.
„MER-VAB was approved by the FDA (Food and Drug Agency) in 2017 for the treatment of patients over 18 years of age. It is registered for the treatment of urinary tract infections including pyelonephritis. Due to increasing multidrug resistance, the drug should only be used against sensitive bacteria”
- Titles of Figure 1 and Figure 2 are wrong. Change it as “chemical structure of meropenem
or vaborbactam.”
Thank you for your attention. The names of the figures have been changed to:
“Fig.1 The 2D (two-dimensional) and 3D (three-dimensional) chemical structure of vaborbactam”.
„Fig. 2 The 2D (two-dimensional) and 3D (three-dimensional) chemical structure of meropenem”.
In addition, the patterns were changed (drawn in a special program) and 3D structure was added.
- In Line 169, the reference “30” does not contain the results mentioned in the paragraph.
The true reference is “Sabet, M., Tarazi, Z., Rubio-Aparicio, D., Nolan, T. G., Parkinson, J.,
Lomovskaya, O., ... & Griffith, D. C. (2018). Activity of Simulated Human Dosage Regimens
of Meropenem and Vaborbactam against Carbapenem-Resistant Enterobacteriacea e in an
In Vitro Hollow-Fiber Model. Antimicrobial agents and chemotherapy, 62(2), 10-1128.”
Check and correct.
Thank you. Our oversight. The citation has been changed.
- Line 164. Change the title. “Pharmacokinetic properties” is more suitable.
Done. Thank you for the suggestion.
- Line 186 and Line 187. How did you correlate the development of resistance to its chemical structure and pharmacokinetics? The connection between this sentence and the paragraph is not clear.
Thank you for your attention. The sentence has been removed to avoid confusion.
- Line 195. Rephrase the sentence. It is too complex to understand.
The complex sentence has been split into two. Now it actually seems clearer and easier to understand.
“Data indicate that the addition of vaborbactam at a dose of 8 mg/lL results in a reduction of the MIC by as much as 8-64 times. This dose of vaborbactam also reduces the MIC of meropenem to ≤ 8mg/mL for all strains”.
- Line 241. Not “MEM-VAB”. It should be “MER-VAB”.
It has been corrected.
- Line 286. Correct the abbreviation.
It has been corrected.
- Line 484. Add the corresponding reference after this sentence.
The reference has been added.
- Line 176 and Line 275. Correct “8hin” as “8h in”.
Done.
As recommended, the work was read and revised by a native speaker.
Thank you very much for your valuable comments. We hope that the corrections we have made are satisfactory to you.
Best regards,
Anna Duda-Madej
Round 2
Reviewer 1 Report
Comments and Suggestions for Authors
The authors haven’t sufficiently improved the quality of the manuscript. Please reject the manuscript for the integrity of the Antibiotics.
The manuscript is poorly presented. Figure 1 and 2, it’s pointless to include the 3D structures of the antibiotics. Figure 3 is also presented in an unprofessional manner.
It’s hard to follow the manuscript. For example, “Most common types of infections among patients in group MER-VAB vs BAT were: 333 BSI (14 vs 8), cUTI/AP (12 vs 4), HABP/VABP (4 vs 1) and cIAI (2 vs 2). K. pneumoniae and 334 E. coli were most frequently isolated pathogens of which CRE pathogens in both groups 335 represented respectively 71.9% vs 93.3% of isolates. In mCRE-MITT population MER-VAB 336 obtained higher efficiency in overall clinical success equally in EOT (end of treatment) and 337 TOC (test of cure).” The manuscript is full of uncommon abbreviations, and it looks like just copy and paste from the experiment part of the cited publications. It lacks of novelty.
Comments on the Quality of English LanguageIt's ok
Author Response
Dear Reviewer,
Thank you very much for re-reviewing our Review Article.
We have corrected the unreadable fragment. It now reads as follows:
„Patients were divided into two groups: those receiving MER-VAB therapy and those receiving BAT. Infections that were most prevalent among patients in both groups were BSI, cUTI/AP, HABP/VABP, and cIAI. K. pneumoniae and E. coli were the most frequently isolated pathogens. CRE pathogens in both groups represented respectively 71.9% and 93.3% of the isolates. MER-VAB was more effective in overall clinical success in both EOT (end of treatment) and TOC (test of cure) in the mCRE-MITT population.”
We have changed Figure 3 to Table 1 to increase the clarity of the message.
Table 1. Scheme of MER-VAB action on individual types of resistance in microorganisms.
Class according to Ambler |
Effect of the vaborbactam |
β-lactamases |
class A |
Inhibition |
Serine carbapenemases: KPC, NMC-A, SME-2 |
ESBL: SHV-2, PER-1 |
||
Narrow spectrum: TEM-1, TEM-2 |
||
class B |
Lack of inhibition |
Metalo-β-lactamases: VIM, NDM-1 |
class C |
Inhibition |
Cephalosporinases: AmpC, P99, ACT-1 |
class D |
Lack of inhibition |
OXA-48 |
In addition, we have removed the 3D structure of meropenem and vaborbactam from Figures 1 and 2.
In addition, we have added a few more expansions of some abbreviations that are not normally used, due to their common knowledge. These are more explanations than expansions, but perhaps they will meet with your approval. Other abbreviations we used were explained when they first appeared, as required. Repeating these explanations would make the Manuscript severely unreadable. We use a lot of names of enzymes, antibiotics, diseases in our review, and only for for expert in this field our review should be clear and easy to understand.
We would also like to point out that the Manuscript is a summary of the results of studies conducted to date using the MER-VAB combination. It is intended to reach that group of readers who 1) study the mechanisms of bacterial resistance and 2) are involved in the therapy of patients with multidrug-resistant infections. Through this article, we want to strengthen public awareness related to the ever-growing problem of multidrug resistance. The message of this Manuscript is also to show that we should not lose hope and look for new associations. Strains such as KPC represent a deadly danger in the hospital environment. It is worth realizing that when resistance to colistin, the antibiotic of "last resort," is determined, an alternative is possible. An alternative in the form of MER-VAB, which can become effective in eradicating the resistant strain.
Our Article follows the definition of Review article is a scientific article that summarizes the current state of knowledge in a given research area. It integrates and interprets the previous results of original scientific research, but does not necessarily contain original research results. In our opinion, we have followed this definition.
In addition, we disagree with the objection"pasting the experimental part" in our review. We have scrupulously analyzed the results of the experiments performed and presented them in the most accessible way possible. This can be confirmed by the plagiarism analysis performed, which was certainly done by the journal.
Best regards,
Anna Duda-Madej
Reviewer 3 Report
Comments and Suggestions for Authors
This version of the manuscript can be accepted for the publication.
Author Response
Dear Reviewer,
Thank you very much for your positive review of our corrections
Best regards,
Anna Duda-Madej
Round 3
Reviewer 1 Report
Comments and Suggestions for Authors
It looks good to me, it can be published in current form.